# Isokinetic Strength, Vertical Jump Performance, and Strength Differences in First Line Professional Firefighters Competing in Fire Sport

**DOI:** 10.3390/ijerph18073448

**Published:** 2021-03-26

**Authors:** Petr Miratsky, Tomas Gryc, Lee Cabell, Frantisek Zahalka, Matej Brozka, Matej Varjan, Tomas Maly

**Affiliations:** 1Research Sport Center, Faculty of Physical Education and Sport, Charles University, José Martího 31, 162 52 Prague, Czech Republic; miratsky@ftvs.cuni.cz (P.M.); gryc.tomas@ftvs.cuni.cz (T.G.); zahalka@ftvs.cuni.cz (F.Z.); brozka.matej@hotmail.com (M.B.); matej.varjan@gmail.com (M.V.); 2Department of Health and Human Performance, Texas State University, San Marcon, TX 78666, USA; lee.cabell@gmail.com

**Keywords:** tactical population, performance, fire sport, asymmetries

## Abstract

The purpose of our study was to investigate peak torque (PT) of knee extensors (KE) and knee flexors (KF), bilateral and unilateral strength asymmetries in isokinetic testing and vertical jump height (JH), vertical ground reaction force (VGRF), and force differences (_Δ_VGRF) between legs during different jump tests in professional first-line firefighters (*n* = 15) competing in fire sports disciplines. There was a significant effect of jump type on JH (F_2,44_ = 7.23, *p* < 0.05), VGRF (F_2,44_ = 16.03, *p* < 0.05), and _Δ_VGRF (F_2,44_ = 3.45, *p* < 0.05). Professional firefighters achieved a mean JH of 50.17 cm in the countermovement jump free arms and high PT of KEs (3.15 Nm/kg). No significant differences (*p* > 0.05) and small effect sizes (*d* < 0.3) were found between the legs when PTs were assessed. We found a slightly higher (*d* = 0.53) unilateral strength ratio in non-dominant legs (58.12 ± 10.26%) compared to dominant legs (55.31 ± 7.51%). No effect of laterality was found among limb comparisons, but a higher unilateral isokinetic strength ratio was found in non-dominant legs of firefighters. A high level of strength (PT of KEs > 3 times body weight) and vertical jump performance is comparable to the performance of elite athletic populations.

## 1. Introduction

Firefighting involves rescuing trapped victims, suppressing and controlling fire spread and environmental damage, and lifting property. During these interventions, firefighters risk their own lives and health to help and rescue other people, animals, or property. This occupation is connected and therefore associated with high levels of physical and psychological demands [1] owing to the nature and difficulty of the occupation. In various studies, firefighters are referred to as tactical athletes. This term includes individuals in service occupations with significant physical fitness and performance requirements, such as special police riot units (SWAT) in the United States, law enforcement, emergency responders, and military personnel [2]. Career firefighters often work in coupled shifts (e.g., 24–48 h at a time), during which they have to manage sleep deprivation and disrupted meal times due to the necessity to rapidly respond to emergency calls. Rapid response to emergency situations requires firefighters to quickly transition from a sedentary state to vigorous physical activity [3]. Professional fire brigades, therefore, require some level or means of exercise among their employees in an effort to assure their physical fitness ability to complete job tasks with high physical demands [4].

Low levels of physical fitness can result in overloading firefighters during demanding interventions. Overexertion is currently the leading cause of fatalities (27.1%) and the second leading cause of musculoskeletal injuries (21%) in U.S. firefighters [5]. These musculoskeletal injuries can also be caused by weakness of the affected muscles, imbalance of the strengths of the agonist and antagonist muscles, and poor muscular flexibility. Many countries and their fire brigades, therefore, place appropriate emphasis on and pay attention to the improvement of these important components of firefighters’ physical and professional fitness. Many interventions and fitness programs have already been developed for this purpose [6,7]. Czech firefighters undergo systematic physical training to keep their fitness level up and to pass annual examinations. The training takes place mainly in sports and training facilities of fire brigade authorities and consists of general and specialized exercises. The latter includes fire sports disciplines and exercises with elements of firefighting activities. Professionals who train and compete in the disciplines of fire sports are referred to as fire sport athletes. However, this approach can lead to various overloads and formations of muscle asymmetries or imbalances since repeated one-side load movements appear during practice for fire-sport disciplines such as running and handling the fire ladder (9 kg); collecting, carrying, and working with fire hoses (3–5 kg); and handling the fire extinguisher (8 kg) during the fire relay. Long-term preferred and uncompensated loads on one side of the body may lead to asymmetry and dominance of one limb, which can be a result of pre-existing limb preference (footedness, handedness) [8]. When this situation is repeated over the many years in which an athlete practices repetitive asymmetric loading, some health problems, such as lower back pain, can be incurred [9]. Whether these firefighters (fire sports athletes) are at risk of muscle imbalances and possible injuries resulting from such physical training must therefore be determined.

Muscular strength is an important component of firefighters’ physical fitness because weak muscles may markedly limit athletic or work performance. Firefighters must maintain high levels of muscular strength to lift and carry heavy pieces of equipment to the site of an emergency, climb stairs, and carry victims to safety, often for extended periods or distances [10]. Many benefits of strength training on musculoskeletal system health have been reported [11].

To assess muscle strength in athletes and tactical athletes, tests such as sit-ups, push-ups, curls, standing long jump, hand grip, and plank are often used in practice [12,13]. Such tests require multi-joint movements; therefore, they do not reflect the strength of the isolated muscle group. These tests refer to complex performance, which results in many repetitions, that is, the period of maintenance of resistance (holding positions). Based on the results of these tests, the detailed characteristics of muscular strength (peak strength or torque, strength power or work, rate of force (torque) development, force decay time, and strength differences between limbs) could not be observed and quantified. The levels of strength and strength asymmetries in athletic populations have been studied over the last three decades to establish the level of performance [14] or for injury-related predictions [15].

Comparing muscle groups from bilateral limbs (i.e., right versus left knee flexors) or from agonist and antagonist muscle groups (i.e., knee flexors versus knee extensors) may reveal a potential weakness that can predispose athletes to injury [16,17]. Increased risk of injury is associated with antagonist/agonist (flexor/extensor) strength ratio lower than 50%—knee flexors (hamstrings) produce lower than 50% of strength of knee extensors (quadriceps). From a clinical significance perspective, the higher flexor/extensor strength ratio the better, but it is commonly accepted that an isokinetic *H:Q* ratio of 60% or greater is desirable in rehabilitation [18,19]. Asymmetrical movements or unilateral loading of muscle parts may occur during firefighting work and, owing to standard firefighting activities performed, as part of interventions or training such as carrying loads, charged hose advance, rolled hose lift and move, Keizer sled and demolition, tasks with ladders, rescuing victims, and working with and carrying tools. Therefore, if these activities occur over a long period repetitively and without compensatory activities, they can lead to the development of various degrees and modes of functional asymmetries [20]. The risk of sustaining tissue damage was reported to increase when the load exerted on a given tissue exceeded its tolerance, e.g., in unilateral activities [21]. However, the above-mentioned studies investigated the physical performance of firefighters, and no study has reported bilateral and unilateral strength deficits in firefighters. Although asymmetrical strength across the body has been linked to various pathological conditions [22], relatively little research is currently being conducted to identify these deficits in special occupations such as firefighting. A wide range of tests to assess muscle power and isokinetic strength performance were used in tactical populations, such as firefighters (first line firefighters and fire department officers), but to our knowledge, no peer-review evidence exists of bilateral and unilateral strength characteristics for the assessment of the risk of injury. Therefore, the present study aimed to investigate the strength performance and effect of limb dominance (bilateral strength asymmetries) and hamstring-to-quadriceps strength ratio (ipsilateral strength ratio) in first line professional firefighters competing in fire sport and compare to active service firefighters (control group). We hypothesized significant differences in vertical jump performance among three different jumps, higher strength values in favor of dominant limb, and higher bilateral isokinetic strength ratio in knee flexors compared to knee extensors. Moreover, we hypothesized significant better physical performance in favor of fire sports athletes group compared to control.

## 2. Materials and Methods

### 2.1. Study Design

A cross-sectional study design was used. We explained the study to all the participants before commencing the study. All the firefighters were informed of the benefits and risks of this study and read and signed an approved consent form. The research was approved by the ethics committee of the Faculty of Physical Education and Sport, Charles University (Czech Republic), under approval No. 205/2020. Research was conducted in accordance with the ethical standards of the Declaration of Helsinki and ethical standards in sport and exercise science research [23].

### 2.2. Participants

Thirty-one male Czech professional first-line city firefighters volunteered to participate in this study. Firefighters were divided into two groups: fire sports athletes group (FSA: *n* = 15) and control group (CG: *n* = 16). Fire sports group subjects (age, 31.13 ± 7.10 years; body height, 183.74 ± 4.31 cm; body mass, 83.23 ± 5.60 kg; body mass index, 24.70 ± 1.33 kg.m^−2^) were in active service and competed in fire sports at national and international championship competitions. In addition to the standard physical training, the participants also trained individually at least five times per week with strength and conditioning programs that included muscular power, speed, and specific exercises for technique development in fire sport disciplines. The control group subjects (age, 33.94 ± 8.98 years; body height, 181.06 ± 5.78 cm; body mass, 80.84 ± 6.13 kg; body mass index, 24.68 ± 1.76 kg.m^−2^) were in active service and undergoing standard physical training. Standard physical training is aimed to increase or maintain physical fitness level as part of regular service (1 per 24 h in 3 days) and includes treadmill and trail running, swimming, team games, and weight training. To eliminate potential confounders as much as possible, the following eligibility exclusion criteria were identified in the research: knee surgery due to injury through entire career, any strenuous physical activity before measurement (48 h), being in fire service last two days, and sleeping over last night before testing.

### 2.3. Anthropometric Procedure

Body height was measured using a digital stadiometer (SECA 242, Hamburg, Germany), and body mass was measured using a digital scale (SECA 769). Body composition variables were assessed using a bioelectrical impedance analysis (Tanita MC-980MA, Tanita Corporation, Tokyo, Japan). The following dependent variables were included: relative values of fat mass (body fat) and fat-free mass. All the participants were tested within a one-week period in the morning before breakfast between 8:00 and 9:30 a.m. to maintain the same testing conditions. The participants were instructed to adhere to the following guidelines before the testing: no food or drink within six hours, no strenuous exercise within 12 h, and no alcohol consumption within 48 h before the test.

### 2.4. Muscle Strength

The concentric muscle strength of the lower limbs was assessed using a Cybex Humac Norm isokinetic dynamometer (Cybex NORM, Humac, Stoughton, MA, USA). The following dependant variables were evaluated during concentric contraction at an angular velocity of 60°.s^−1^: maximum peak muscle torque of the knee extensors (PTE) and flexors (PTF) in the dominant (DL) and non-dominant (NL) legs, bilateral strength ratio between the knee extensors (*Q*:*Q*) and flexors (*H*:*H*), and unilateral ratio (hamstring-to-quadriceps ratio (*H*:*Q*)) for both limbs. Limb dominance was defined by determining with which leg each participant preferred to kick (kicking leg) [24]. The lower limbs were evaluated in random order. Torque was gravity-corrected, and dynamometer calibration was performed in accordance with the manufacturer’s instructions. After five submaximal warm-up repetitions, the subjects performed three repetitions with maximum effort. Visual feedback and verbal stimulation were provided during the testing.

Two Kistler 8611 force plates (Kistler Group, Winterthur, Switzerland) at sampling frequency of 1000 Hz were used to measure and evaluate muscular power, force distribution, and jump height. Force curves were recorded using BioWare 5.4.3.0 software (Kistler Group, Winterthur, Switzerland) and for additional data processing, Matlab R2020b software was used (MathWorks, Natick, MA, USA). Three jump types were performed in the same order for each participant to evaluate the explosive strength and symmetry involvement of the lower limbs, namely counter-movement jump free arms (CMJ_FA_), counter-movement jump (CMJ), and squat jump (SQJ). For CMJ_FA_, subjects were instructed to perform a CMJ with arm swing during the execution of the jump (i.e., hands were free to move). For CMJ, subjects started from the upright standing position with their hands on their hips (i.e., without arm swing); they were then instructed to flex their knees (approximately 90°) as quick as possible and then jump as high as possible in the ensuing concentric phase. For SQJ, subjects started from the upright standing position with their hands on their hips, they were then instructed to flex their knees and hold a predetermined knee position (90°), and the experimenter then counted out for 3 s. On the count of three, the subject was instructed to jump as high as possible without performing any countermovement before the execution of the jump [25]. The best of the three attempts based on the height of each jump type was selected for data processing. Changes of body velocity, displacement, and jump height were calculated by impulse–momentum method [26] from common force curve from both platforms. Separate force curves from each platform were used for force and impulse of force evaluation of DL and NL. The assessed parameters were jump height (cm), maximal vertical ground reaction force (VGRF) of the DL and NL, their differences (_Δ_VGRF), and vertical ground reaction force impulse at the take-off phase (_Δ_VGRF_t−o_) differences. Before measurements, each participant warmed up for 15 min consisting of 10 min of cycling on an ergometer at 1.2 W.kg^−1^/80–90 rpm (Excalibur, Lode, NL), two sets of forward lunges with 10 repetitions, and two sets half squats with 10 repetitions.

### 2.5. Statistical Analyses

The data were processed using descriptive statistics (mean, standard deviation, and 95% confidence interval). The assumptions were met before proceeding with the parametric statistical analyses. Analysis of variance (ANOVA) was used to detect differences between the dependent variables. Multiple comparisons of the averages of the dependent variables were performed using the Bonferroni post-hoc test. To compare the observed indicators between the DL and the NL, we used a paired *t* test to compare the means of two dependent samples. The effect size in the ANOVA model was assessed using the “partial Eta square” coefficient (*η*_p_^2^). Coefficient values <0.010 indicated a small effect; 0.011–0.059, small-to-medium effect; 0.060–0.138, medium-to-large effect; and >0.139, large effect [27]. For comparison among two groups (FSA vs. CG), we used *t* test for independent samples (with normal data distribution) or Mann–Whitney test (nonparametric test). Moreover, the effect size between the DL and the NL was assessed using Cohen’s *d* coefficient of effect size. The coefficient value was assessed as follows: *d* = 0.20, small effect; *d* = 0.50, medium effect; and *d* = 0.80, large effect [28]. The *p*-value was set at 0.05 in all the statistical analyses. Statistical analyses were performed using IBM SPSS v 21 (Statistical Package for Social Sciences, Inc., Chicago, IL, USA, 2012).

## 3. Results

The results revealed the significant effect of jump type on jump height (JH), VGRF, and _Δ_VGRF (Table 1). Post hoc analyses revealed significant differences in JH parameters (*p* < 0.01) between CMJ_FA_ (50.17 ± 7.84 cm) and CMJ (43.64 ± 5.87 cm) and SQJ (40.99 ± 6.57 cm). VGRF was significantly lower in SQJ (21.09 ± 2.35 N.kg^−1^) than in CMJ (26.10 ± 3.43 N.kg^−1^) and CMJFA (25.51 ± 1.67 N.kg^−1^). In addition, _Δ_VGRF was significantly higher in CMJ (8.9 ± 6.74%) than in SQJ (4.15 ± 3.54%). No significant differences were found between the different jump tasks in _Δ_VGRF_t−o_ (*p* > 0.05; Table 2) and between the DL and the NL in VGRF production (Table 2).

The isokinetic strength performance is presented in Table 3. No significant differences and small effect sizes were found between the DL and the NL. The professional firefighters with unilateral asymmetries achieved slightly greater but not significantly higher unilateral strength ratio (*H*:*Q*) in the NL (58.12 ± 10.26%) than in the DL (55.31 ± 7.51%). The unilateral difference between the lower limbs reached a medium effect size. The bilateral strength asymmetries did not reveal differences between the extensors and flexors (*p* ≥ 0.05, *d* = small).

The significantly higher jump height, isokinetic strength of knee extensors, flexors on dominant and non-dominant limb have been detected in FSA group compare to CG (Figure 1 and Figure 2). Conversely, no significant differences were revealed when comparing bilateral isokinetic strength asymmetries and unilateral strength ratio (Figure 3 and Figure 4).

## 4. Discussion

The purpose of this study was to investigate the levels of isokinetic and vertical jump performance of first line professional firefighters competing in fire sport. Strength performance and the effects of limb dominance (bilateral strength asymmetries) and hamstring-to-quadriceps strength ratio (ipsilateral strength ratio) were observed, and possible bilateral and unilateral asymmetries in muscle strength and bilateral strength performance engagement were assessed in firefighters during their physical training, in addition to their general physical training, and special training, which included the disciplines of fire sports.

Scientific data are lacking regarding the explosive strength characteristics in professional firefighters. One of the reasons for this is that the laboratory tests used to measure the muscle strength in firefighters are complicated, expensive, and time consuming [29]. Cornell et al. [30] reported JH in the CMJ (42.40 ± 6.52 cm) test in 32 U.S. volunteer firefighters. Moreover, the authors monitored jump performance during 14- and 38-week firefighter training academy programs. The authors reported non-significant changes after 14 (41.05 ± 5.44 cm) and 38 weeks (41.23 ± 5.74 cm). Perroni et al. [31] evaluated Italian firefighter recruits (*n* = 16 lower JH in CMJ (28.9 ± 5.0 cm)) unlike our study (43.64 ± 5.87 cm). Dawes et al. [32] tested a sample of two groups of full-time highway patrol officers. One of the tests was also an explosive force evaluated using the electrical contact operated system Just Jump (ProBotics Inc., Huntsville, AL, USA). The JH test was performed for group 1 (45.74 ± 7.46 cm); and the CMJ test for group 2 (48.23 ± 7.57 cm). Frio Marins et al. [33] reported JH in CMJ (36.2 ± 3.8 cm) and in SQJ (29.8 ± 3.5 cm) of federal highway police officers (*n* = 13). We suggest that the higher performance levels in the CMJ and SQJ tests of the firefighters participating in this study are the result of the systematic training process for fire sport disciplines. In addition to standard first line firefighters, fire sports athletes practice physical training every day. Fire sport disciplines are commonly included to special physical training in Czech Republic first-line firefighters, however, firefighters (fire sports athletes) participating in this study compete in fire sport on national and international level which requires high intensity special training. Fire sports consist of four disciplines: 100-m hurdles, 4 × 100-m hurdles, climbing the tower, and fire attacks. Each discipline requires high strength and high level of rate force development of the lower limbs to reach maximum speed, whole-body strength to carry equipment, and adequate technique to carry and manipulate the equipment at maximum speed. Therefore, training consists mostly of strength and speed exercises as well as specific technique training for fire sports disciplines.

Higher CMJ_FA_ values (61.5 ± 7.1 cm) than those in our study (50.17 ± 7.84 cm) were reported in fire academy trainees [34], but contact mat was used and the JH results were calculated from flying time, which overestimated the results by up to 2–3 cm [26] when compared to JH calculated from the force characteristics (acceleration phase), used in our study. Significant differences in JH between the different jump tasks were found in our study. The highest jumps were recorded in CMJ_FA_ (50.17 ± 7.84 cm), followed by the CMJ and SQJ (43.64 ± 5.87 cm vs. 40.99 ± 6.57 cm). We believe this is due to greater impulse being generated with the incorporation of arm swing during CMJ_FA_ when compared to CMJ as well as a lack of stretch-shortening cycle being incorporated in SQJ. On the other hand, in the VGRF parameter, the best results were recorded in CMJ (26.10 ± 3.43 N.kg^−1^), followed by CMJ_FA_ and SQJ (25.51 ± 1.67 N.kg^−1^ and 21.09 ± 2.35 N.kg^−1^, respectively). We suggest that this is due to the nature of the load when performing work duties in which the upper limbs are “excluded” from locomotor movements, such as carrying loads, holding and handling hoses, pulling various materials and firefighting tools, or rescuing people, or activities (firefighting, rescue, and demolition) that often have “static” isometric positions such as squatting, holding a hose, or moving up a ladder.

We found significant differences between jumps in terms of VGRF force between the lower limbs (Table 2). The VGRF production by our firefighters in the CMJ test was similar (26.10 ± 3.43 N.kg^−1^) when compared with elite Brazilian soccer players (26.78 ± 2.83 N.kg^−1^) [35]. A higher VGRF in the SQJ test (27.41 N.kg^−1^) was published for elite Spanish soccer players by Centeno-Prada et al. [36] when compared with our participants in the SQJ test (21.09 ± 2.35 N.kg^−1^). The highest _Δ_VGRF asymmetry was found in CMJ (8.90 ± 6.74%). Conversely, the lowest value was detected in SQJ (4.15 ± 3.54%). To our best knowledge, no data from professional firefighters are available for comparison. Our results are in line with those of Ferreira et al. [35], who reported ground reaction force asymmetry in elite soccer players (20 years old; 8.63 ± 3.81%). Maly et al. [20] reported bilateral ground reaction force asymmetry (8.65 ± 4.88%) in CMJ in untrained boys (age = 14.7 ± 0.3 years). In comparison with the present study, they found similar _Δ_VGRF asymmetries in CMJ_FA_ (8.11 ± 5.50%) and SQJ (8.54 ± 4.44%), which is controversial when compared with our study in professional firefighters. No significant functional asymmetries or imbalances in the lower limbs were reported in this study for professional firefighters. These positive findings can be considered unexpected, often due to the one-sided load during the practice of fire sport disciplines mentioned above.

The results of the present study showed that professional firefighters generated high PT in both the DL (3.15 ± 0.36 N.m.kg^−1^) and the NL (3.08 ± 0.37 N.m.kg^−1^). The strength performance levels of our firefighters were higher than the published results in other studies. Noh et al. [37] reported data from South Korean national firefighters (DL: 2.48 ± 0.52 N.m.kg^−1^, NL: 2.39 ± 0.50 N.m.kg^−1^). Gerstner et al. [38] reported lower strength performance (2.04 ± 0.46 N.m.kg^−1^) in adult career firefighters. High performance differences between our firefighters and those in the report by Gerstner et al. [38] should be due to (i) the different level of performance (fire sport athletes vs. career firefighters), (ii) absolute body weight and BMI index differences (BW: 83.23 ± 5.60 kg vs. 109.30 ± 20.57 kg, BMI: 24.70 ± 1.33 kg.m^−2^ vs. 33.24 ± 4.95 kg.m^−2^), and (iii) age difference between the compared groups (31.13 ± 7.10 years vs. 36.90 ± 6.87 years). Our results pertaining to performance PT (3.06 ± 0.44 N.m.kg^−1^) are comparable with those from elite junior soccer players (I. Belgian league). Menzel et al. [39] reported higher knee extensor PT values in professional soccer players (*n* = 46) for the DL (3.36 ± 0.51 N.m.kg^−1^) and NL (3.43 ± 0.57 N.m.kg^−1^). Maly et al. [40] reported similar values in height level kickboxers as compared with our results (DL: 3.04 ± 0.54 N.m.kg^−1^, NL: 3.13 ± 0.52 N.m.kg^−1^). No significant differences were found in the PT of the extensors and flexors between the DL and NL (Table 3). The PT of the KF in our study (DL: 1.74 ± 0.31 N.m.kg^−1^, NL: 1.79 ± 0.37 N.m.kg^−1^) was higher than that of national Korean firefighters (DL: 1.28 ± 0.26 N.m.kg^−1^, NL: 1.25 ± 0.33 N.m.kg^−1^).

This contrast can be explained by the advanced knee flexors specific load adaptation that fire sport events require. A comparison of the unilateral strength ratio showed a slightly higher *H*:*Q* ratio in the NL (58.12 ± 10.26%) than in the DL (55.31 ± 7.51%), which had a medium effect size. The physically active population did not show a statistically significant difference between the bilateral *H*:*Q* ratio between the limbs in the other studies [41]. The typical *H*:*Q* ratio of a healthy knee ranges from 50% to 80%, depending on the knee angle and angular velocity [19]. Our *H*:*Q* values were similar to those in young active population [20]. The authors also confirmed a significantly higher *H*:*Q* ratio for active populations (football players) than for non-active subjects. The level of bilateral differences was <10% for both KE and KF. In athletic populations [42], greater asymmetry (>10%) in KE increases the risk of musculoskeletal injuries by 16-fold and ligament and meniscus injuries by up to 28-fold. Moreover, bilateral strength asymmetry in KF (>10%) increases the risk of injury by 12-fold. As some firefighters had higher bilateral differences (>10%), more attention should be paid to compensation for strength asymmetries by tailored strength programs. Although asymmetrical actions are a part of the occupation, the loads that they are asked to carry may not be significant to develop asymmetries, or lack of task-specificity (which asymmetries have been shown to be dependent upon) could be the potential reason.

The special sample of first-line city firefighters (fire sports athletes) showed a high level of strength performance with insignificant force differences between the lower limbs. For this reason, it is necessary to treat them as “athletes” to maintain or increase physical condition. Since this population would be considered well trained, the loads of the carries, ladders, etc., may not be significant enough to create asymmetries compared to the training these individuals do on a regular basis. This may highlight the benefit of a well-rounded training program to minimize development of asymmetries in these populations. Targeted and planned sports training using the principle of periodization should be part of the regular physical training of professional firefighters. The physical fitness of professional firefighters should also be regularly assessed using appropriate diagnostic tools, such as vertical jump performance and isokinetic strength testing. The results presented in our study can also be helpful to other researchers for the purpose of further comparative studies with other groups of tactical athletes.

Although presented study aimed on determination of isokinetic and explosive strength, cardiorespiratory fitness is also an important component of good health-related physical fitness. A low level of cardiovascular fitness is associated with risk factors of cardiovascular diseases [43], such as hypertension, dyslipidaemia, and obesity, which are leading cause of on-duty death or disability [44] in firefighters. Obesity is associated with a sedentary job and lack of movement, irregular sleeping, unbalanced and irregular nutrition, and effect of 24 h shifts as factors threatening first-line firefighters [45]. Therefore, it is important to have regular physical training, especially aimed at cardiovascular fitness and muscle strength, and verify their level regularly using appropriate test batteries, such as physical examination for members of Czech fire brigades to control their minimal physical standards. Level of cardiorespiratory fitness, body composition components, and flexibility of firefighters competing in fire sport could be an interesting purpose of future research.

Despite the presented results, the study had several limitations. These included testing at a low angular velocity (60°.s^−1^) with concentric muscle contraction and considering just the highest PT without referring specific joint angle across range of motion. Unilateral load could negatively affect the lower and upper limbs and trunk muscles in terms of unilateral and bilateral symmetries. Future research should therefore focus on functional and structural asymmetries, not only of the lower limbs, but also of the upper limbs and torso where an increased degree of asymmetry can be expected owing to the nature of the load. Furthermore, our participants who competed in fire sports represented a special population classified among tactical athletes; therefore, our results are valid only for similar firefighter units. In addition, the research was conducted under controlled laboratory conditions, with no stress factors occurring during the firefighter events. Future research should also include firefighters who take advantage of exercises with elements of firefighting activities during their training and compete in the Toughest Firefighter Alive. Other similar professions include police riot units, law enforcers, emergency responders, and military personnel belonging to tactical units.

## 5. Conclusions

The present study results showed a high level of explosive strength and isokinetic muscle strength of the knee extensors and flexors in firefighters competing in fire sports. Professional firefighters achieved high strength performances that are comparable with elite athletic populations. Knee extensors can produce high relative muscle strength (three times the body weight), which is also comparable with those in professional athletes. The effect of limb’s dominancy was not statistically confirmed in any of the observed variables. The significantly higher jump height, isokinetic strength of knee extensors, flexors on dominant and non-dominant limb were found in FSA group compared to CG, while no significant differences were revealed when comparing bilateral isokinetic strength asymmetries and unilateral strength ratio. As some firefighters had higher bilateral differences (>10%) or lower unilateral differences (<50%), more attention should be paid to compensate for strength asymmetries using tailored strength programs. Monitoring the levels of strength and strength asymmetries of professional firefighters is important for the optimal health and performance-related fitness and well-being of firefighters. Moreover, our results may be useful to clinical practitioners, i.e., lecturers, strength and conditioning trainers, physiotherapists, physicians, and other clinical professionals working with highly physically and mentally demanding activities such as those involved in the occupation of firefighting.

## Figures and Tables

**Figure 1 ijerph-18-03448-f001:**
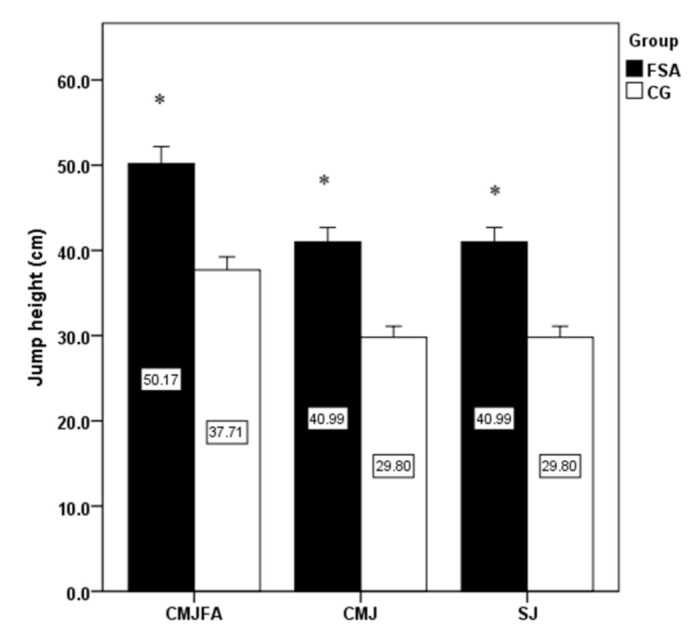
Comparison of jump height differences between groups. Note: CMJFA: countermovement jump with free arms; CMJ: countermovement jump; SJ: squat jump; FSA: fire sports athletes; CG: control group; *: significant difference between groups.

**Figure 2 ijerph-18-03448-f002:**
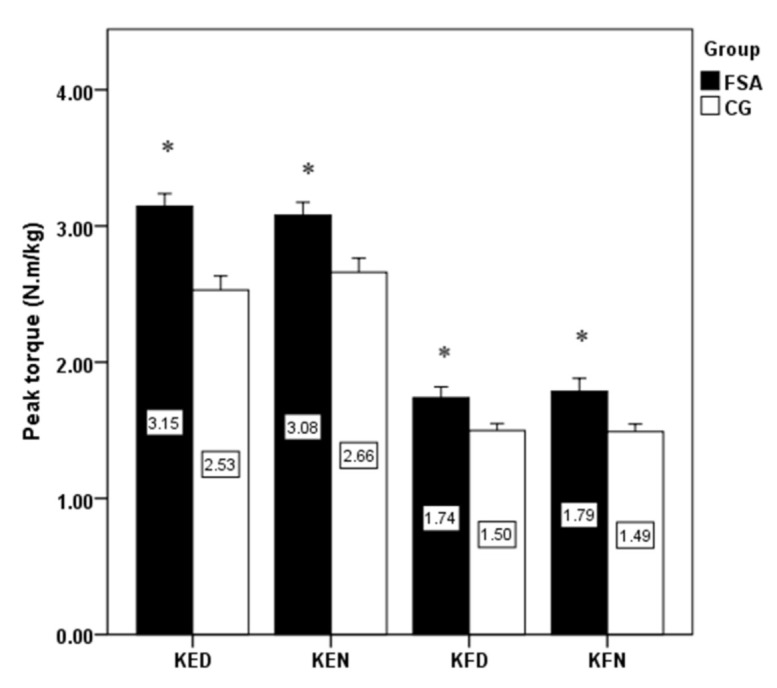
Comparison of peak torque differences between groups and limbs. Note: KED: knee extensors on dominant limb; KEN: knee extensors on non-dominant limb; KFD: knee flexors on dominant limb; KFN: knee flexors on non-dominant limb; FSA: fire sports athletes; CG: control group; *: significant difference between groups.

**Figure 3 ijerph-18-03448-f003:**
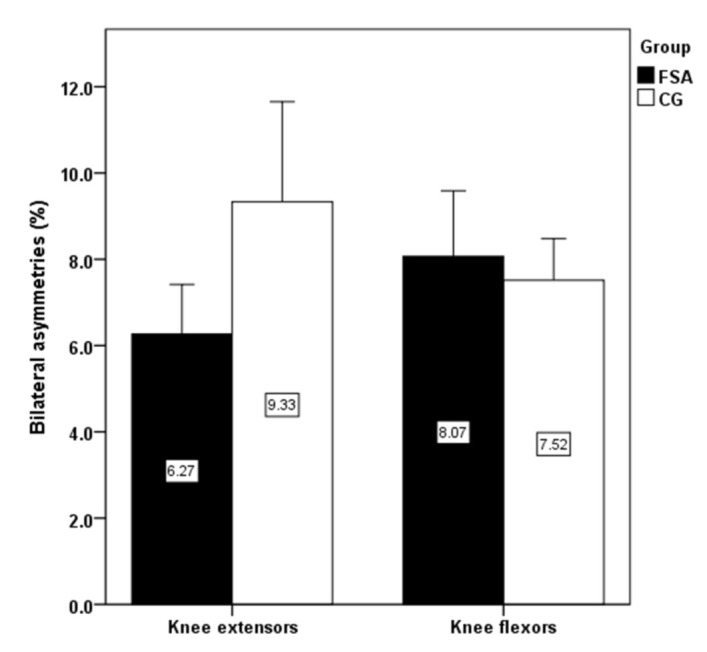
Comparison of bilateral strength differences between groups. Note: FSA: fire sports athletes; CG: control group.

**Figure 4 ijerph-18-03448-f004:**
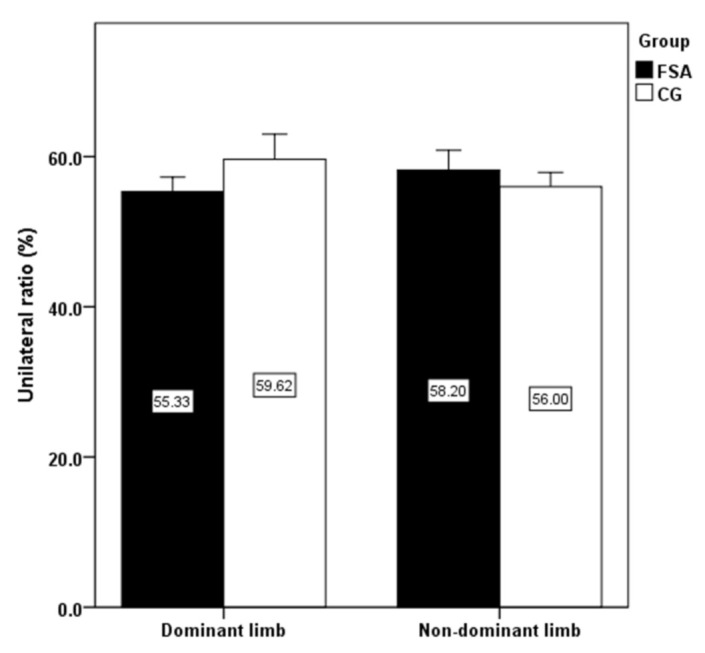
Comparison of unilateral strength ratio differences between groups. Note: FSA: fire sports athletes; CG: control group.

**Table 1 ijerph-18-03448-t001:** Comparison of vertical jump performance among the different jump tests.

Parameters		Descriptive Statistic				ANOVA		Bonferroni’s Post-Hoc Test
		Mean	SD	95% CI Interval		F	*p*	
				Lower	Upper			
JH (cm)	CMJ_FA_	50.17	7.84	45.83	54.51	7.23	0.00	CMJ_FA_ vs. CMJ, SQJ
	CMJ	43.64	5.87	40.39	46.89			
	SQJ	40.99	6.57	37.36	44.63			
VGRF (N.kg^−1^)	CMJ_FA_	25.51	1.67	24.53	26.39	16.03	0.00	CMJ_FA_ vs. SQJ, CMJ vs. SQJ
	CMJ	26.10	3.43	24.23	27.69			
	SQJ	21.09	2.35	19.72	22.56			
_Δ_VGRF (%)	CMJ_FA_	5.50	4.50	3.01	7.99	3.45	0.04	CMJ vs. SQJ
	CMJ	8.90	6.74	5.16	12.63			
	SQJ	4.15	3.54	2.19	6.11			
_Δ_VGRF_t−o_ (%)	CMJ_FA_	4.46	3.40	2.58	6.34	0.20	0.82	
	CMJ	4.50	3.44	2.59	6.40			
	SQJ	3.79	3.50	1.85	5.72			

Note: JH, jump height; CMJFA, countermovement jump free arm; CMJ, countermovement jump; SQJ, squat jump; VGRF, maximal vertical ground reaction force; _Δ_VGRF, limb differences of the maximal vertical ground force production; _Δ_VGRF_t−o_, limb differences in vertical ground force impulse production during the take-off phase; SD, standard deviation.

**Table 2 ijerph-18-03448-t002:** Comparison of VGRF among the dominant and non-dominant limbs (N.kg^−1^).

Jump Test	Dominant Limb		Non-Dominant Limb		*t*	Sig.	*d*	Effect Size
	Mean	SD	Mean	SD				
CMJ_FA_	12.85	0.89	12.66	0.98	1.01	n.s.	0.21	small
CMJ	12.85	1.67	13.24	2.16	−0.91	n.s.	0.20	small
SQJ	10.60	1.47	10.60	1.08	0.30	n.s.	0.00	small

Note: CMJFA, countermovement jumps free arms; CMJ, countermovement jump; SQJ, squat jump; SD, standard deviation; *d*: Cohen’s effect size coefficient; n.s.: non-significant.

**Table 3 ijerph-18-03448-t003:** Isokinetic strength performance and unilateral and bilateral strength ratios.

Variable	Dominant Limb		Non-Dominant Limb		*t*	Sig.	*d*	Effect Size
	Mean	SD	Mean	SD				
KE (Nm/kg)	3.15	0.36	3.08	0.37	1.07	n.s.	0.19	small
KF(Nm/kg)	1.74	0.31	1.79	0.37	−0.86	n.s.	0.15	small
	**Unilateral differences**							
	**Dominant limb**		**Non-dominant limb**					
UR (%)	55.31	7.51	58.12	10.26	−1.30	n.s.	0.53	medium
	**Bilateral differences**							
	**Knee extensors**		**Knee flexors**					
BR (%)	6.27	4.45	8.07	5.89	−1.18	n.s.	0.35	small

Note: KE: Knee extensors; KF: Knee flexors; UR: Unilateral ratio; BR: Bilateral ratio; SD, standard deviation; *d*: Cohen’s effect size coefficient; n.s.: non-significant.

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
