# Peer review of "Isokinetic Strength, Vertical Jump Performance, and Strength Differences in First Line Professional Firefighters Competing in Fire Sport"

_ijerph, 2021, doi:10.3390/ijerph18073448_

Round 1

Reviewer 1 Report

You demonstrated that a high level of strength and power were found in first-line firefighters, which is comparable to elite athletes. No effect of laterality was found among limb comparisons, but a higher unilateral isokinetic strength ratio was found in non-dominant legs of firefighters.

however you have to discuss about the importance of cardio respiratory good physical condition as VO2max and about the specific firemen training 

Author Response

Dear reviewer,

thank you very much for very stimulating and helpful revision. We realized that the submitted work can be improved based on the assessment and incorporation of comments. We appreciate your time and effort in order to provide your revision with aim to improve the submitted manuscript. We believe that the new version has a higher quality and would be interesting for readers. We believe that the published research issue will find a large circle of readers and will also be cited in future articles. Once again, we would like to sincerely thank you for your time, effort, professional and factual comments.

Yours authors.

Response on your comments, as well as justification, or correction you can find in attached file please.

Reviewer 2 Report

This interesting manuscript explores measures of muscle performance (knee extension and flexion) in professional firefighters aged 19-41y.  While interesting, I find several limitations which prevent publication of the manuscript in its current form.  A major concern is that the conclusions stated comparing firefighters to elite athletes may be misleading as the sample size is very small and there is no evidence that the subjects are a random sampling of firefighters that accurately represent this population as a whole.  For example, median firefighter age is 38.6y and 79.5% have a BMI greater than 25 (33.5% with BMI greater than 30) in the United States (Poston J Occup Environ Med 2011).  These are substantially greater than the means in the current study.  Indeed, the authors speak of “tactical athletes” and go to some length describing subjects’ participation in exercise training and international athletic competitions.  The manuscript should be revised to reflect this subgroup of firefighters.  Specific comments appear below.

Abstract:

Line 11: clarify PT indicates peak torque

Line 13: clarify JH refers to Jump height

Line 17-18: References to “high” PT and comparisons to elite athletes should be left to the conclusion portion of the abstract.  Results should be stated objectively without interpretation.

Line 19: misleading conclusion of higher strength ratio in non-dominant leg as p-value as not sufficient to reject the null hypothesis. 

Introduction:

Line 30: the reference to COVID-19 does not appear to contribute meaningfully to the discussion regarding muscle strength.

Lines 33-35 seem a bit redundant

Line 74:  if mulitjoint movements are so crucial to sport/professional performance, why did the authors choose to assess single joint strength?

Line 85: what values of antagonist/agonist muscle asymmetry are associated with increased risk of injury?  From my understanding a knee flexor/extensor strength ration of 60% is within normal bounds.  Therefore, both limbs in the present study appear to display normal antagonist/agonist strength.  What is the clinical significance of modestly greater strength ratio in the non-dominant limb if it still falls within normal bounds? 

Lines 94-103: need to revise to address the specific “fire sport athletes” that are studied in the present paper rather than generalizing to all firefighters.

Methods:

Line 146: need to reframe paper to indicate that concentric muscle strength is assessed

Line 151: why was limb dominance defined by self-report preferred kicking leg.  Are the data different if defined by highest PT?  This may be a more accurate definition of limb dominance.

Line146-154: why is KE and KF peak torque compared for strength ratio?  Did the authors consider comparing KF torque specifically at the knee joint angle in which KE PT was achieved?  It would be interesting to observe the strength ratio across the full ROM in dominant and non-dominant limbs.

Line 160: provide description or references for how each specific jump was performed.

Results:

Line 185: “HJ” should be defined in the methods or at first use.  I believe this is intended to be “JH” as indicated in the abstract.  Please clarify.

Line 186: refers to a “firefighter group.” Were data compared to a non-firefighter group?  This statement is confusing.

Line 200:  Remove “high” this interpretation should be saved for the discussion/conclusion

Discussion:

Lines 211-214:  Remove the journal’s text:  “Authors should discuss the results and how they can be interpreted from the per-211 spective of previous studies and of the working hypotheses. The findings and their implications should be discussed in the broadest context possible. Future research directions may also be highlighted.”

Need to reframe discussion to reflect subpopulation studied; cannot generalize to entire firefighter population.  It would be good to include a brief discussion of the prevalence of obesity and general physical fitness of the firefighter population as a whole.

Author Response

Dear reviewer,

thank you very much for very professional and helpful comments. We realized that the submitted work can be improved based on the assessment and incorporation of comments. We appreciate your time and effort in order to provide your revision with aim to improve the submitted manuscript. We believe that the new version has a higher quality and would be interesting for readers. We believe that the published research issue will find a large circle of readers and will also be cited in future articles. Once again, we would like to sincerely thank you for your time, effort, professional and factual comments.

Yours authors.

Response on your comments, as well as justifications, or corrections you can find in attached file please.

Reviewer 3 Report

General comments: Title of manuscript is somewhat misleading. The main focus is asymmetries whereas title reads as though there may be comparisons to different populations. Consider revising to reflect analyses performed.

Overall there are some questions related to purpose and outcome from this study. It may be useful to differentiate between high-performing and low-performing individuals (as determined by researchers and one of the assessments that were performed) to make comparisons in that manner if sample size can be increased. The main finding was a lack of asymmetries that were observed, yet the main focus is discussing how asymmetries may be detrimental or prevalent related to this population. If this is one of the first studies looking at this population (or asymmetries in tactical populations), where did the notion that this population is likely to develop asymmetries? Although asymmetrical actions are a part of the occupation, the loads that they are asked to carry may not be significant to develop asymmetries or due to lack of the task-specificity (which asymmetries have been shown to be dependent upon) could be the potential reason. I would recommend including some of this information within the discussion to further support this possibility for lack of findings pertaining to asymmetries.

Define all abbreviations first before using within manuscript rather than just in the abstract and then not defining them within other areas of manuscript.

Lines 30-31 - Unsure how COVID-19 relates to relevance of data collected or purpose of study.

Lines 61-63 - Why would this type of training lead to muscle imbalances? Provide examples of training or competitions that would lead you to make this statement for this population.

Lines 69-71 - There are several research studies as well as reviews of literature that emphasize the importance of muscular strength to athletic performance/activities such as jumping, sprinting and change of direction. To say that the importance of strength relative to physical performance in sports may be misguided. Please review work of Suchomel, The importance of muscular strength in athletic performance, for research related to importance of strength on physical performance

Lines 99-101 - You state there is no peer-reviewed evidence in this population for injury risk, however injury risk is not a part of this study at all. This research

Line 105 - Power was not measured as a part of this study, only jump height. Ground reaction forces were measured for each limb during jumps but that is not a measure, nor indicator of power since it does not include a velocity component.

Lines 129-130 - confused as to the sleep requirements/restrictions in place to be included in study. Please re-word.

Methods-Muscle Strength:

-How was jump height calculated?

-Was the data filtered or raw data used for analysis?

-What depth was used for squat jump?

-How long did participant stay in squat before jumping?

-Were all jumps performed in the same order or were they randomized?

-How were countermovement jumps performed? Hands on hips or hands on head or another manner?

-Is asymmetry determined by subtracting dominant versus non-dominant limb for each participant?

-What threshold is being used to determine take-off?

Line 185-HJ was not defined previously. Also recommend changing all "HJ" abbreviations to JH to reflect more common abbreviations within the literature. Jump height is abbreviated as JH in abstract.

Line 188 - SQJ and SJ are used interchangeably for squat jump. Please revise to keep consistent.

Line 200- revise to high peak torque

Table 2. Recommend changing title to
"Comparison of power assessment level among different jump tests"

Lines 211-214 - remove text referring to instructions for authors.

Lines 243-247 - Was a contact mat used in this study of Kistler force plates? Are reported values of 2-3 cm from this research study or previous research (28)? Please report both jump height calculations in results if data is coming from this study.

Line 250 - suggest removing "good whole body coordination" as reason for greater jump heights being observed. It is likely due to greater net impulse being generated with the incorporation of arm swing which is commonly observed as well as lack of stretch-shortening cycle being incorporated in the squat jump.

Line 313-316 Great point about the population observed. This may likely be a reason for lack on asymmetries observed. Since this population would be considered well trained, the loads of the carries, ladders, etc. may not be significant enough to create asymmetries compared the training these individuals do on a regular basis. This may highlight the benefit of a well-rounded training program to minimize development of asymmetries in these populations. 

Author Response

(The authors gave the same response as above.)

Round 2

Reviewer 1 Report

This article has now been much improved in the specific objective measure of this work to characterize the physical qualities of elite firefighters in a competitive approach.

Author Response

Thank you very much for yours comments and your final positive conclusion.

Yours authors.

Reviewer 2 Report

We appreciate the authors’ detailed response to suggestions and criticisms from all reviewers.  The authors are commended for so clearly addressing these concerns.  Unfortunately, I believe there was some misunderstanding regarding a few of my previous comments.  Overall, I find this to be a relatively interesting descriptive manuscript which is novel in the specific population observed.  However, the utility of the information is a bit lost on this reviewer.

Abstract:

Line 21:  The “higher (d=0.53) unilateral strength ratio…” statement is misleading because the p-value is not sufficient to reject the null hypothesis.  Although the authors not an effect size of 0.53, the p-value indicates this difference is not statistically significant.  Therefore, the authors cannot conclude there is evidence to support a higher strength ratio in the non-dominant limb as the null hypothesis is accepted.  Furthermore, the “medium” effect size noted is < 3 percentage points greater in the non-dominant limb.  The physiological relevance of this difference (even if the difference was significant) is likely miniscule. 

Introduction:

Line 85: I appreciate the authors’ response to my previous comment about antagonist/agonist strength ratio.  Inclusion of the Ghena et al, Orchard et al, and Kong et al. data indicating a desirable ration of 60% would bolster the authors’ argument that strength ratios are an important outcome and provide a numerical reference to put the reported 55 and 58% strength ratios into perspective.  Please briefly include this.

Methods:

Limb dominance: Thank you for examining the data as suggested.  It appears that defining by peak torque did not have a strong relationship with self-report kicking strategy.  I still wonder whether the observed relationships would be different if limb dominance was defined by highest peak torque.  However, the authors' point is well taken that this may different between agonist and antagonist muscle groups.

Author Response

Dear editor(s) and reviewer(s),

we really appreciate once again for your comments, valuable and interesting notes with aim to increase the quality of submitted paper. We have considered carefully all of them and almost all we incorporated to manuscript together with justification. The present version has a higher quality for readers and hope that after your review will be possible to publish in International Journal of Environmental Research and Public Health.

Yours authors.

In the following lines we would like to describe the scope of changes and reactions to the reviewers' recommendations. The described changes apply to the word document in which the text changes are tracked (Tracked version document please)

Reviewer 3 Report

General comments: Thank you for taking the time to make recommended changes and addressing each one. Overall I think the manuscript is improved greatly. I feel as though this manuscript would be greatly improved if there were some type of comparison to other firefighters, possibly those that are not participating in competitive fire sport. I truly feel as though this is such a unique subset of the tactical and firefighter population that it may not represent the population. It is still worthy of adding to the literature identifying this population as what would be considered very well trained individuals. It would be interesting to see how these individuals compare to others as well on some more occupation specific tasks to demonstrate that the increased physical fitness in these individuals also translates to occupation performance.

Lines 161-163 - change to : Force curves were recorded using BioWare 5.4.3.0 software (Kistler Group, Winterthur, Switzerland) and for additional data processing Matlab R2020b software was used (MathWorks, USA).

Within discussion, several formatting errors with regards to references. Lines 238, 241, 243, 247, 281, 285, 287, 298, 299, 301, 307, 309

Lines 293-294 - Spacing issues

Author Response

Dear  reviewer,

we really appreciate once again for your comments, valuable and interesting notes with aim to increase the quality of submitted paper. We have considered carefully all of them and almost all we incorporated to manuscript together with justification. The present version has a higher quality for readers and hope that after your review will be possible to publish in International Journal of Environmental Research and Public Health.

Yours authors.

In the following lines we would like to describe the scope of changes and reactions to the reviewers' recommendations. The described changes apply to the word document in which the text changes are tracked (Tracked version document please)
